# PRIVDISTIL: A UNIFIED FRAMEWORK FOR ACCURATE AND DIFFERENTIALLY PRIVATE MODEL COMPRESSION

## ABSTRACT

Privacy has emerged as a paramount concern in the development and deployment of language models. While over-parameterized models deliver exceptional performance, their deployment on resource-constrained devices (such as mobile or embedded systems) remains challenging. Model compression techniques are widely adopted to address this, yet they introduce additional privacy risks. Although differential privacy (DP) provides rigorous theoretical safeguards against data leakage, the noise injection required during compression often severely degrades model utility. Balancing high performance with strong privacy guarantees in compressed models thus remains a critical open challenge. In this work, we introduce PRIVDISTIL, a DP-aware model compression framework that redesigns the compression pipeline to preserve utility while protecting sensitive data. PRIVDISTIL begins by training a domain classifier via DP-SGD on a hybrid dataset of public and private samples, thereby identifying public data most aligned with the private domain. It then performs model compression exclusively on these selected public samples, followed by fine-tuning the compressed model on the private dataset using DP-SGD. By shifting the compression burden to public data, PRIVDISTIL minimizes noise requirements and boosts training stability. Extensive experiments show that PRIVDISTIL consistently surpasses state-of-the-art DP compression methods across diverse datasets and architectures, delivering an average accuracy gain of over 3% on GLUE benchmarks under a strict privacy budget of $\varepsilon = 1$.

## 1 INTRODUCTION

Language models are powerful deep learning systems capable of producing text that closely resembles human communication. They are typically based on the Transformer architecture (Vaswani et al., 2017) and often contain billions of parameters. The emergence of models such as BERT (Devlin et al., 2019; Liu et al., 2019) and the GPT series (Radford et al., 2019; Brown et al., 2020) has revolutionized *natural language processing* (NLP), ushering in a new paradigm for training deep neural networks. These models are first pre-trained on vast and diverse public datasets, and then fine-tuned with smaller, domain-specific datasets that may include sensitive user data. Despite their strong performance, over-parameterized language models pose challenges when deployed in resource-constrained environments (e.g., mobile or embedded devices).

Moreover, deploying smaller, compressed models *locally* on mobile or embedded devices can offer substantial advantages over hosting large models via external APIs. First, local inference preserves user privacy by keeping sensitive data on-device rather than transmitting it to remote servers. Second, on-device processing reduces latency and reliance on a stable internet connection, improving user experience in bandwidth-constrained or offline scenarios. Finally, running efficient compressed models locally lowers operational costs and energy consumption, making them especially appealing for large-scale or real-time applications.

While increasing model size tends to boost performance, multiple studies have shown that over-parameterized models are prone to leaking private information contained in their training data (Shokri et al., 2017; Carlini et al., 2021; 2019). In practice, the downstream datasets for special-

ized tasks can include highly sensitive information such as medical records or financial transactions. *Differential privacy* (DP) (Dwork et al., 2006) provides a principled defense against such leaks and has been widely applied in NLP and computer vision (Yu et al., 2022; Mehta et al., 2022; Li et al., 2022; De et al., 2022). By injecting controlled noise, a differentially private algorithm guarantees that any single example in the training set contributes only marginally to the final model, limiting the ability of adversaries to infer its presence. Formally, this is expressed as $(\varepsilon, \delta)$-DP, where smaller values of $(\varepsilon, \delta)$ imply stronger privacy guarantees. The most prominent approach for training neural networks under DP is Differentially Private Stochastic Gradient Descent (DP-SGD) (Abadi et al., 2016). In DP-SGD, each individual gradient is clipped to a fixed norm, and a noise term proportional to the clipping norm is added to the sum of the gradients. Although this process provides privacy guarantees, the injected noise can substantially degrade model accuracy, especially for large, deep architectures (Abadi et al., 2016; Tramèr & Boneh, 2021; Shen et al., 2021). This trade-off is even more pronounced when compressing language models for practical deployment, as both fine-tuning and compression must be privatized, compounding the noise impact. As a result, managing the trade-off among privacy, efficiency, and accuracy in this context is particularly challenging.

Our central insight is that in resource-limited deep learning workflows, two stages—fine-tuning and compression—necessitate differential privacy guarantees, and each suffers from noise-induced performance degradation. By reversing their order and performing compression with public data first, we reduce noise injection to a single step. Nevertheless, this simple reordering may cause the compressed model to miss important task-specific knowledge. To address this, we introduce the PRIVDISTIL framework, which employs a data selection strategy to identify the most relevant public samples for the private domain.

Concretely, PRIVDISTIL uses DP-SGD to train a domain classifier on a mixed dataset of public and private data. The trained classifier then evaluates all public samples, selecting those with the highest similarity to the private domain for knowledge distillation, thus circumventing further noise addition during compression. We further refine the selected data via clustering and high-uncertainty sampling to ensure that it retains both diversity and relevance. Finally, after compressing the model on this curated public dataset, we apply a DP-SGD fine-tuning step on the private dataset to adapt the compressed model to the target task. In this reconfigured workflow, only the relatively small domain classifier and the final DP fine-tuning stage incur privacy costs. As domain gaps between public and private data are typically significant, the domain classifier converges swiftly with a lower privacy overhead. Accordingly, PRIVDISTIL strategically decreases the total DP noise burden across the compression and fine-tuning pipeline, resulting in higher utility for private tasks.

We implement PRIVDISTIL using language models, including BERT$_{\text{BASE}}$, BERT$_{\text{LARGE}}$, and DistilBERT, and evaluate on GLUE tasks such as SST-2, QQP, QNLI, and MNLI. Following the setup in (Mireshghallah et al., 2022), our experiments demonstrate that PRIVDISTIL consistently outperforms the strongest existing approach at the same privacy budgets and compression rates, significantly improving the privacy-utility trade-off in compressed language models. For instance, compressing BERT$_{\text{BASE}}$ into DistilBERT at $\varepsilon = 1$ increases mean accuracy from 76.1% with the prior state-of-the-art to 79.2%—an improvement exceeding 3% across four datasets. Notably, such a margin is substantial in DP learning, where even 1% gains are considered meaningful (Wei et al., 2022; Fu et al., 2023). In another case, on SST-2 with a privacy budget of $\varepsilon = 4$, our method narrows the performance gap between private and non-private training from 91.3% (non-private baseline) and 82.7% (SOTA) to 89.2%, a near 4× reduction.

**Contributions**. Our key contributions are outlined below:

• *Innovative Framework*. We present PRIVDISTIL, a differentially private model compression approach that achieves significant improvements in the privacy-utility trade-off.

• *Advanced Techniques*. We propose Adjusted Importance Sampling (AIS), which jointly considers data quality and domain relevance to bolster private fine-tuning and potentially improve other scenarios involving public data.

• *Extensive Empirical Studies*. We evaluate PRIVDISTIL across diverse datasets and settings, demonstrating that it preserves high utility under strong privacy guarantees.

## 2 PRELIMINARIES

### 2.1 MODEL COMPRESSION

Pre-trained language models can be compressed through various techniques such as low-rank approximation (Ma et al., 2019; Lan et al., 2020), weight sharing (Dehghani et al., 2019; Lan et al., 2020), pruning (McCarley, 2019; Fan et al., 2020; Elbayad et al., 2019; Gordon et al., 2020; Hou et al., 2020; Cui et al., 2019), quantization (Shen et al., 2020; Zafrir et al., 2019), or knowledge distillation (Hinton et al., 2015; Sanh et al., 2019; Jiao et al., 2020; Wang et al., 2020; Turc et al., 2019; Liu et al., 2020; Sun et al., 2020). In this paper, we primarily focus on knowledge distillation.

*Knowledge Distillation.* Knowledge Distillation (KD) transfers the knowledge of a large "teacher" network $T$ to a smaller "student" network $S$, training the student to mimic the teacher's behaviors. Let $f^T$ and $f^S$ denote the teacher and student behavior functions, which transform inputs into meaningful representations. In pre-trained language models, these functions typically correspond to outputs of certain Transformer layers (Vaswani et al., 2017). Each Transformer layer consists of multi-head attention (MHA) and a feed-forward network (FFN). For distillation, any of these outputs—e.g., MHA/FFN activations or attention matrices—can serve as the behavior function.

Formally, KD aims to minimize:

$$\mathcal{L}_{\text{KD}} = \sum_{x \in \mathcal{X}} L\big(f^S(x), f^T(x)\big), \tag{1}$$

where $L(\cdot)$ denotes a loss function that measures discrepancy between the teacher and student networks, $x$ is the input text, and $\mathcal{X}$ is the training dataset.

### 2.2 DIFFERENTIALLY PRIVATE MACHINE LEARNING

*Differential privacy* (Dwork et al., 2006) (DP) is a formal framework for safeguarding individual-level information in a dataset when conducting statistical analyses or releasing model outputs. It ensures that outputs from a randomized mechanism remain nearly indistinguishable whether or not any single data record is included in the input, thus protecting individuals' privacy while still permitting broad insights into the data distribution.

**DP-SGD**. In privacy-preserving deep learning, *differentially private stochastic gradient descent* (DP-SGD) (Abadi et al., 2016) is the de facto standard. DP-SGD modifies the gradient estimation of traditional SGD by clipping each sample's gradient to a fixed norm and adding Gaussian noise proportional to this clipping norm before summing. This mechanism obscures any single example's contribution to the overall gradient. The total privacy budget for DP-SGD is derived by tracking the per-iteration privacy cost (under certain $(\varepsilon, \delta)$ constraints) and then applying composition and subsampling amplification techniques (Bun & Steinke, 2016; Dwork et al., 2010; 2014) across iterations.

## 3 THE PRIVDISTIL FRAMEWORK

### 3.1 PROBLEM FORMULATION

We seek to transform a pre-trained LLM into a smaller, privacy-preserving model suitable for specific tasks. Combining model compression and differential privacy, our objective is a compact language model (LM) that adheres to data privacy while remaining within a constrained size budget. The key challenge is to produce a compressed LM that rivals the performance of its larger counterpart without overshooting the size limit or compromising privacy.

Formally, let $\varepsilon > 0, \delta > 0$ be privacy parameters. We start with a public pre-trained LLM (initial parameters $\theta_{LLM}(0)$), a private dataset $\mathcal{D}_{priv}$ for a target downstream task, and a compression factor $\gamma$. Let $|LLM|$ denote the number of parameters in the LLM. We aim to derive a compressed LM such that: (i) $|LM| \leq \gamma \cdot |LLM|$, and (ii) the final parameters of the compressed model, $\theta_{LM}(t)$, satisfy $(\varepsilon, \delta)$-differential privacy with respect to $\mathcal{D}_{priv}$. Any method may use LLM in the process, provided that the final weights $\theta_{LM}(t)$ remain DP-compliant.

We measure compression quality by comparing the accuracy of an LM (trained under $(\varepsilon, \delta)$-DP) to that of the original LLM (also $(\varepsilon, \delta)$-DP) on the same downstream task. This comparison reveals how much performance loss is introduced by compression under private training constraints. Notably, we do *not* compare to non-private baselines: our goal is to find compression algorithms that preserve as much of the differentially private LLM's utility as possible.

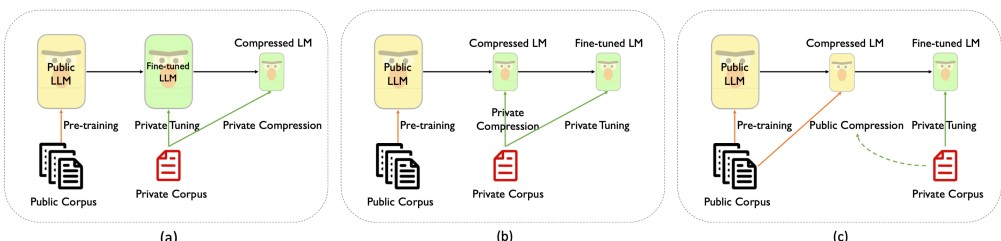

(a)                                (b)                                (c)

Figure 1: Three frameworks for LLM compression.

### 3.2 STRAWMAN SOLUTIONS

**Strawman Solution I: Private Fine-tuning then Private Compression**. Mireshghallah et al. (2022) propose DPKD, shown in Figure 1(a), where both fine-tuning and compression run under DP. This approach, however, may suffer markedly from DP noise. Prior works indicate that large models, when trained with DP-SGD, tend to perform worse than smaller ones (Shen et al., 2021). Consequently, directly fine-tuning the larger LLM under DP-SGD can substantially degrade the final model's accuracy.

**Strawman Solution II: Private Compression then Private Fine-tuning**. A naive alternative is to reverse the order of compression and fine-tuning, as illustrated in Figure 1(b). By compressing first, the subsequent DP noise applies to a smaller student model rather than the full LLM.

However, prior research reveals that simply applying DP-SGD to the knowledge distillation process does not yield a differentially private student (Mireshghallah et al., 2022). The crux lies in the dependence of $f^T(x)$ on the entire dataset, which cannot be sanitized via straightforward clipping and noise addition (Mireshghallah et al., 2022). As a result, the arrangement in Figure 1(a) remains the only certified DP-compliant framework among these two strawman solutions. Moreover, our experiments confirm that a more carefully orchestrated approach (as in Figure 1(c)) outperforms this reversed pipeline (detailed in Section 4.6).

### 3.3 OUR PROPOSAL

**Intuition**. To overcome the limitations in prior approaches, our main idea is to make the compression process *public*, as depicted in Figure 1(c). Specifically, we employ a public corpus during the knowledge distillation phase, eliminating the need to include private data—which in turn removes the necessity for DP noise during compression. However, if the compression relies exclusively on public data, it lacks awareness of the private domain and thus mainly distills *general* knowledge rather than the *task-specific* knowledge required. Crucially, enabling a compact model to learn the input-output mappings of a larger model is only feasible when the input space is sufficiently constrained, a condition not met if we rely on general-purpose public data alone. Consequently, such a purely public distillation step typically underperforms (Section 4.2).

To address this gap, we propose PRIVDISTIL—a *data-centric* solution that strategically selects data from the public corpus based on its similarity and usefulness to the private domain. Because the selected corpus remains public and contains no sensitive information, it can be safely used for knowledge distillation. In this manner, the distillation process inherently accounts for the private task's data distribution without requiring privacy measures.

**Overview**. Figure 2 illustrates the workflow of PRIVDISTIL. To harness public data for beneficial knowledge distillation tailored to a private task, PRIVDISTIL operates through four key steps:

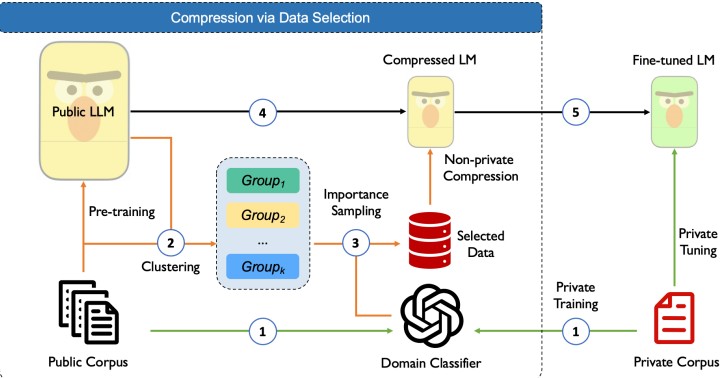

Figure 2: Overview of the PRIVDISTIL framework.

• *Step 1: Domain Classifier Training.* We begin by constructing a binary domain classifier that distinguishes between private and public data. This classifier is trained on a mixture of public and private samples and thus requires DP noise to protect private data. Once trained, it infers which public examples are more likely to mirror the private domain.

• *Step 2: Public Corpus Clustering.* In addition to relevance, we also need diversity in the selected public samples to ensure robust generalization. We cluster the public corpus based on teacher-model embeddings, thereby leveraging the teacher model's representational capabilities to partition the data into coherent groups.

• *Step 3: Adjusted Importance Sampling.* We introduce *adjusted importance sampling* (AIS) to integrate both diversity and domain relevance. Within each cluster, we select samples with the highest estimated relevance to the private task (as indicated by the domain classifier) until we achieve a desired selection size.

• *Step 4: Public Knowledge Distillation.* Because the chosen public samples are non-sensitive, knowledge distillation can proceed without DP constraints. We follow a procedure akin to Distil-BERT, transferring knowledge from the teacher to a reduced-layer student model. Large batches, dynamic masking, and gradient accumulation ensure efficient training on our curated public dataset. The student model emerges as a compact yet highly effective proxy for the larger model.

Finally, after public knowledge distillation, we *fine-tune* the compressed model on the private dataset under DP constraints, completing the PRIVDISTIL pipeline.

### 3.4 DESIGN DETAILS

**Domain Classifier Training.** Since our primary objective is to select public data that effectively benefits the private task, we start by identifying samples closely aligned with the private data distribution. Formally, suppose we have a small set of private samples $x'_1, x'_2, \ldots, x'_n$ drawn from a distribution $p$, and a large public dataset $x_1, x_2, \ldots, x_N$ drawn from a distribution $q$. Our goal is to extract $m$ samples ($m \ll N$) from the public corpus that best approximate the private distribution $p$.

Building on the data cleansing techniques from GPT-3, The Pile, and PaLM (Brown et al., 2020; Chowdhery et al., 2022; Gao et al., 2021), we train a binary domain classifier $f : \mathcal{X} \to [0, 1]$ on a mixed dataset of private (labeled 0) and public (labeled 1) examples. Because private data are used in this training, we apply DP-SGD to maintain privacy. Once trained, $f(x_i)$ provides an estimate of how likely a public sample $x_i$ is to come from the private domain. We then select public samples that receive the highest $f(x_i)$ scores, effectively picking those most similar to the private distribution.

**Public Corpus Clustering.** Beyond domain relevance, we also want diversity in our selected samples to improve generalization. Toward this end, we cluster the entire public dataset $\mathcal{D}_{\text{pub}}$ using embeddings produced by the teacher model. Specifically, we define an embedding function

$$e : \mathcal{D}_{\text{pub}} \to \mathbb{R}^d,$$

which transforms each public sample into a $d$-dimensional representation. We then apply the $k$-means algorithm on these embeddings to form $k$ clusters:

$$C_k = \arg\min \sum_{i=1}^{k} \sum_{x \in C_i} \|e(x) - \mu_i\|^2,$$

where $C_i$ is the $i$-th cluster and $\mu_i$ is its centroid. Intuitively, increasing $k$ yields more granular clusters and potentially higher diversity. However, choosing an overly large $k$ does not always improve results, as shown in Section 4.5.

**Adjusted Importance Sampling.** To balance relevance and diversity in our selected data, we introduce *adjusted importance sampling* (AIS). Let $\mathcal{S} = \{s_1, s_2, \ldots, s_m\}$ be the set of $m$ chosen samples. We iterate over each of the $k$ clusters, selecting the sample that maximizes $f(x)$, i.e.,

$$s_i = \arg\max_{x \in C_k} f(x),$$

where $f(x)$ is the trained domain classifier's output. We repeat this process (one sample at a time per cluster) until we reach $m$ samples, ensuring that each cluster is proportionately represented. This strikes a balance between transferring knowledge relevant to the private domain and maintaining sufficient diversity for test-time generalization.

**Public Knowledge Distillation.** Because the selected samples come entirely from a *public* corpus and therefore contain no sensitive information, we can distill the teacher model's knowledge without adding DP noise. Following a procedure similar to DistilBERT (Sanh et al., 2019), we first construct a student model by mirroring the teacher's architecture while removing the token-type embeddings and pooler, and halving the number of Transformer layers. Since the hidden dimensions of teacher and student match, we initialize the student by copying every other layer from the teacher to preserve representational consistency. We then train the student on the selected public samples using knowledge distillation with large batch sizes, gradient accumulation, and dynamic masking. This non-private distillation produces a significantly more compact model while retaining strong performance. Finally, we apply DP-SGD fine-tuning on the private dataset to adapt the compressed model to the downstream task under formal privacy guarantees.

**Overall Algorithm.** The complete procedure of PRIVDISTIL is summarized in Algorithm 1 in Appendix A. It details the five-stage pipeline, including domain classifier training, clustering, adjusted importance sampling, public knowledge distillation, and final private fine-tuning under DP-SGD.

## 4 EVALUATION

### 4.1 EXPERIMENTAL SETUP

We follow the setup in (Mireshghallah et al., 2022), our primary baseline. We use pre-trained teachers: $\text{BERT}_{\text{BASE}}$[1] (12 layers, hidden size 768) and $\text{BERT}_{\text{LARGE}}$[2] (24 layers, hidden size 1,024). The student is a 6-layer DistilBERT variant, initialized from the teacher per (Sanh et al., 2019), with matching hidden size. For the domain classifier, we fine-tune pre-trained Distilled-GPT2[3] on mixed private and public data. All checkpoints are from Hugging Face. We evaluate on five datasets: OpenWebText (Gokaslan et al., 2019) (large unlabeled web text, used as public corpus); and private tasks from GLUE (Wang et al., 2019): MNLI (393k sentence pairs for inference), QNLI (100k+ question-paragraph pairs for entailment), QQP (400k+ question pairs for duplicates), and SST2 (67k movie reviews for sentiment).

We compute privacy budgets via numerical composition (Gopi et al., 2021), evaluating $(\varepsilon = 4, \delta = 1/|D|)$ and $(\varepsilon = 1, \delta = 1/(10|D|))$, with clipping threshold $C = 0.1$. Total budget covers domain classifier training ($\varepsilon \approx 0.3$) and fine-tuning; hyperparameter tuning is excluded per standard practice (Mireshghallah et al., 2022; Tramer & Boneh, 2020). We provide the detailed configuration of domain classifier training, model compression, and fine-tuning in Appendix B. This includes how we handle multi-sentence samples for MNLI, QNLI, and QQP, our choice of training epochs, and the use of large-batch DP-SGD with Opacus.

---

[1]https://huggingface.co/bert-base-uncased
[2]https://huggingface.co/bert-large-uncased
[3]https://huggingface.co/distilgpt2

Table 1: Comparison of the performance of PRIVDISTIL, DPKD, and vanilla fine-tuning across various teacher models and under different privacy budgets.

| Privacy Budget | Model | Teacher | Training | MNLI | QNLI | QQP | SST2 | Avg |
|---|---|---|---|---|---|---|---|---|
| $\varepsilon = 1$ | BERT$_{BASE}$ | - | Finetune | 74.8 | 85.6 | 82.1 | 86.8 | 82.3 |
| | DistilBERT (BASE/LARGE) | - | Finetune | 66.9 | 81.0 | 78.3 | 79.6 | 76.5 |
| | | BERT$_{BASE}$ | DPKD | 67.5 | 80.1 | 78.4 | 78.5 | 76.1 |
| | | | PRIVDISTIL (Ours) | **69.0** | **80.2** | **81.8** | **85.6** | **79.2** |
| | | BERT$_{LARGE}$ | DPKD | 67.6 | 80.1 | 78.0 | 78.0 | 75.9 |
| | | | PRIVDISTIL (Ours) | **68.3** | **80.1** | **80.9** | **82.3** | **77.9** |
| $\varepsilon = 4$ | BERT$_{BASE}$ | - | Finetune | 77.8 | 87.8 | 84.7 | 90.5 | 85.2 |
| | DistilBERT (BASE/LARGE) | - | Finetune | 71.7 | 83.2 | 82.4 | 82.7 | 80.0 |
| | | BERT$_{BASE}$ | DPKD | 72.8 | 83.0 | 82.6 | 82.7 | 80.3 |
| | | | PRIVDISTIL (Ours) | **73.3** | **83.3** | **84.8** | **89.2** | **82.7** |
| | | BERT$_{LARGE}$ | DPKD | 72.4 | **83.1** | 81.1 | 81.5 | 79.5 |
| | | | PRIVDISTIL (Ours) | **72.8** | 82.8 | **83.9** | **85.4** | **81.2** |

## 4.2 EFFECTIVENESS OF PRIVDISTIL

We empirically validate PRIVDISTIL's effectiveness against the SOTA baseline DPKD (Mireshghallah et al., 2022) across privacy budgets and compression ratios. Table 1 reports accuracies on private tasks. PRIVDISTIL consistently improves accuracy across settings. For $\varepsilon = 1$ with BERT$_{BASE}$, it boosts average accuracy from 76.1% (DPKD) to 79.4% (+3.3%), a substantial gain for GLUE tasks (Wang et al., 2019). Gains are smaller at $\varepsilon = 4$ (+2.4%), likely due to reduced DP noise impact in baselines, limiting our noise-mitigation benefits. With BERT$_{LARGE}$ at $\varepsilon = 1$, PRIVDISTIL advances from 75.9% to 77.7%, confirming efficacy on larger models. Compared to direct fine-tuning of the uncompressed teacher ($\varepsilon = 1$, BERT$_{BASE}$: 82.3%), PRIVDISTIL keeps the drop within 3%, remaining stable across budgets (2.9% at $\varepsilon = 1$; 2.5% at $\varepsilon = 4$), indicating that compression introduces minimal DP overhead. PRIVDISTIL also outperforms direct student fine-tuning in all cases, underscoring the value of knowledge distillation.

Efficiency gains are shown in Table 2: BERT$_{BASE}$ speeds up by 1.97×, and BERT$_{LARGE}$ by 3.77×. Notably, for the same budget and task, BERT$_{LARGE}$ may underperform BERT$_{BASE}$ due to its higher compression ratio (4:1 vs. 2:1), complicating knowledge transfer. We use a consistent distillation algorithm for fair comparison; stronger algorithms are beyond this scope.

Table 2: Efficiency summary.

| System | # Params | Speedup |
|---|---|---|
| BERT$_{BASE}$ (Teacher) | 109.5M | ×1.0 |
| BERT$_{LARGE}$ (Teacher) | 335.1M | ×1.0 |
| DistilBERT$_{BASE}$ | 67.0M | ×1.97 |
| DistilBERT$_{LARGE}$ | 108.4M | ×3.77 |

The efficiency of the compressed model is summarized in Table 2. In detail, BERT$_{BASE}$ is sped up by 1.97 times, while BERT$_{LARGE}$ is sped up by 3.77 times.

It can be observed that, for the same privacy budget and the same private task, BERT$_{LARGE}$ could underperform compared to BERT$_{BASE}$. This can be attributed to the fact that the compression ratio for BERT$_{LARGE}$ is 4:1 (24:6), which is higher than the ratio for BERT$_{BASE}$ at 2:1 (12:6). A higher compression ratio can make it more challenging for the student model to assimilate crucial weights, necessitating a stronger knowledge distillation algorithm. We note that in this study, we maintained a consistent knowledge distillation algorithm to ensure fair comparison, and enhancing the knowledge distillation algorithm is beyond the scope of our research.

Table 3: The impact of data selection method.

| Data Selection | MNLI | QNLI | QQP | SST2 | Avg |
|---|---|---|---|---|---|
| Full data | 73.0 | 82.8 | 84.3 | 87.7 | 81.9 |
| Random Selection | 66.4 | 74.8 | 81.5 | 80.3 | 75.7 |
| Domain Classifier | 71.3 | 81.8 | 84.4 | 88.3 | 81.5 |
| AIS (Ours) | **73.3** | **83.3** | **84.8** | **89.2** | **82.7** |

## 4.3 NECESSITY OF DATA SELECTION

**Setup**. We set the privacy budget to $\varepsilon = 4$, fix the teacher model to $\text{BERT}_{\text{BASE}}$, and maintain a compression ratio of 0.5. Furthermore, we compare the data selection methods with the utilization of the full *OpenWebText* dataset, which is labeled as "Full data" in the table, while other data selection methods adhere to a number of 500k selected data samples to ensure a fair comparison.

**Observations**. Table 3 illustrates the impact of different data selection methods on the performance of the PRIVDISTIL framework. We can observe that utilizing the full dataset only moderately outperforms random selection: while random selection yields an average accuracy of 75.7%, employing the full data achieves an average accuracy of 81.9%, improving the performance by just 6.2%. This could be attributed to the fact that larger models often require substantial data volumes to enhance generalization. In contrast, smaller models, being constrained by their capacity, derive limited benefit from an increase in data size, but can significantly benefit from accessing high-quality data.

Furthermore, we observe that employing the domain classifier notably enhances the performance of the PRIVDISTIL framework. Utilizing only the domain classifier can attain an average accuracy of 81.5%, which is an average improvement of 5.8% compared to random selection. This underscores that elevated transferability, achieved through higher relevance to the private data, can significantly benefit the PRIVDISTIL framework. Additionally, it can be observed that implementing the proposed AIS can further boost PRIVDISTIL's performance. With the incorporation of AIS, the average accuracy can be further elevated to 82.7%, marking an average improvement of 1.2% compared to using only the domain classifier. This highlights the crucial necessity of maintaining a balance between transferability and test-time generalization.

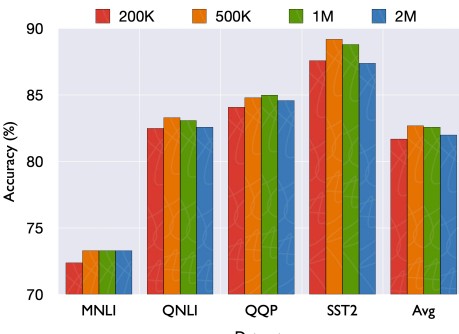

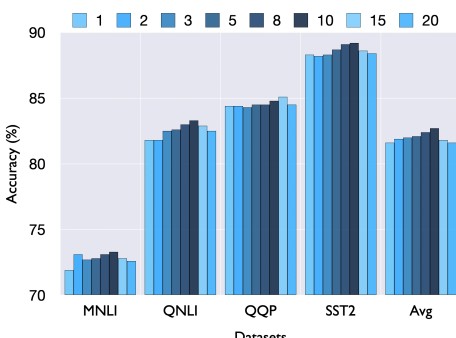

Figure 3: Impact of the number of selected data.

Figure 4: Impact of the number of clusters.

## 4.4 IMPACT OF NUMBER OF SELECTED DATA

From Table 3, it is evident that utilizing the full data does not yield sufficiently good results. Consequently, it is imperative to also manage the quantity of selected data to attain optimal performance. A larger size diminishes the impact of the data selection method but enhances generalization ability. Therefore, the number of selected data also serves to balance generalization ability and transferability — the latter being facilitated by the data selection method through increasing data relevance to the private data domain. In this experiment, we set the privacy budget to $\varepsilon = 4$, fix the teacher model to $\text{BERT}_{\text{BASE}}$, and maintain a compression ratio of 0.5.

**Observations**. Figure 3 illustrates how the number of selected data influences the performance of the PRIVDISTIL framework. We observe that, with the exception of the QQP dataset, PRIVDISTIL typically achieves peak performance when the data selection size is 500k. As a result, we set the default number of the selected data to 500k in our experiments. Although the optimal data size for the QQP dataset is near 1M, as shown in Figure 3, its performance diminishes as it further increases to 2M. This underscores the crucial necessity of maintaining a balance between transferability and test-time generalization.

## 4.5 IMPACT OF NUMBER OF CLUSTERS

Recall that the number of clusters, $k$, acts as a hyperparameter in the proposed PRIVDISTIL, serving to balance generalization ability and transferability. In this section, we explore the impact of $k$ on the performance of PRIVDISTIL. In this experiment, we set the privacy budget to $\varepsilon = 4$, fix the teacher model to $\text{BERT}_{\text{BASE}}$, and maintain a compression ratio of 0.5.

**Observations**. Figure 4 sheds light on the relationship between the number of clusters and the performance of the PRIVDISTIL framework. We observe that, most datasets, except QQP, exhibit optimal performance with the number of clusters is set to 10. As a result, we set the default number of the clusters to 10 in our experiments. QQP deviates from this trend, peaking at approximately 15 clusters, as indicated in Figure 4, yet it experiences a performance decline upon extending to 20 clusters. This pattern underscores the pivotal role of striking a judicious balance between transferability and test-time generalization.

Table 4: Comparing with the strawman solution.

| Training | MNLI | QNLI | QQP | SST2 | Avg |
|---|---|---|---|---|---|
| DPKD | 72.8 | 83.0 | 82.6 | 82.7 | 80.3 |
| KD+FT | 69.7 | 83.2 | 81.9 | 87.4 | 80.6 |
| PRIVDISTIL (Ours) | **73.3** | **83.3** | **84.8** | **89.2** | **82.7** |

## 4.6 COMPARING WITH STRAWMAN SOLUTION II

Although existing research has demonstrated that the Strawman Solution II fails to create student models that satisfy DP due to the lack of protection on the outputs of $f^T(x)$ in Equation 1 (as discussed in Section 3.2), we temporarily ignore this privacy leakage and examine how such a framework performs. We denote the Strawman Solution II as *KD+FT*. In this experiment, we set the privacy budget to $\varepsilon = 4$ (without considering the privacy leakage in the middle), fix the teacher model to $\text{BERT}_{\text{BASE}}$, and maintain a compression ratio of 0.5.

**Observations**. Table 4 showcases the results achieved by DPKD, KD+FT, and PRIVDISTIL. It can be observed that KD+FT consistently outperforms DPKD, elevating the average accuracy from 80.3% (achieved by DPKD) to 80.6%. This could be attributed to two reasons: firstly, KD+FT typically alters the order of compression and fine-tuning, which reduces the DP impact (as discussed in Section 3.2). Secondly, KD+FT introduces additional privacy leakage compared to DPKD, thus resulting in an unfair comparison. Conversely, PRIVDISTIL consistently achieves higher performance than KD+FT, with an average accuracy of 82.7%. This underscores that the proposed PRIVDISTIL surpasses the strawman solution, even in the face of an unfair comparison.

## 5 CONCLUSION

In this work, we introduced PRIVDISTIL, a framework designed to enhance the privacy-utility trade-off for large language models under private fine-tuning. By strategically reorganizing the fine-tuning and compression stages, and employing the Adjusted Importance Sampling (AIS) mechanism for data selection, PRIVDISTIL enables more effective knowledge distillation without compromising privacy. Empirical results across prominent models and diverse datasets validate the efficacy of PRIVDISTIL, demonstrating its potential for building private, efficient, and accurate language models in real-world applications.

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

# A ALGORITHM

Algorithm 1 summarizes the complete PRIVDISTIL pipeline. Given a pre-trained public LLM, a public dataset $\mathcal{D}_{pub}$, and a private dataset $\mathcal{D}_{priv}$, along with user-defined parameters (clusters $k$, samples $m$, training epochs $R_{priv}$, $R_{pub}$, $R_{dc}$, and privacy budget $(\varepsilon, \delta)$), the procedure outputs a compressed model that satisfies $(\varepsilon, \delta)$-DP. The process is divided into five stages: (1) building and training a domain classifier with a portion of the privacy budget, (2) clustering the public corpus, (3) selecting public data via adjusted importance sampling, (4) conducting public knowledge distillation on the selected samples, and finally (5) privately fine-tuning the model using DP-SGD on $\mathcal{D}_{priv}$.

## A.1 ALGORITHM ANALYSIS

**Model Utility.** In PRIVDISTIL, only two components are subject to DP noise, as highlighted by the green models in Figure 2: the domain classifier and the final fine-tuned student model. Crucially, the domain classifier is only used for data selection and does not directly affect the model's final accuracy, since it lies outside the main pre-training, compression, and fine-tuning pipeline. Hence, whereas two large models were previously perturbed by DP (see Figure 1), we now reduce the noise impact to essentially one main component (the student model). Moreover, the large model previously subjected to DP noise (i.e., the teacher) is replaced by a much smaller domain classifier. Since prior studies indicate that larger models suffer more from DP-induced noise (Shen et al., 2021), our approach can significantly decrease the detrimental effect on utility compared to earlier frameworks.

**Privacy Budget.** Although PRIVDISTIL introduces a second DP-protected component (the domain classifier), this classifier is notably smaller and simpler than the teacher model it replaces. Moreover, such a simple model often converges faster under DP-SGD and thus accrues a lower overall privacy cost. This phenomenon arises because each DP-SGD iteration expends part of the privacy budget, meaning fewer required iterations translate to less total privacy expenditure. Additionally, public and private datasets typically differ substantially in real-world scenarios, making the domain-classification task relatively straightforward. Consequently, the domain classifier can often achieve satisfactory performance with a minimal privacy budget. In cases where the private and public datasets closely resemble each other, the classifier need not be highly precise—again allowing a modest budget. In both scenarios, PRIVDISTIL yields lower total privacy costs than the previous DP-KD framework, where the large teacher model also consumed privacy budget.

**Privacy Guarantee.** Recall that the input to PRIVDISTIL comprises a public pre-trained LLM, a public corpus $\mathcal{D}_{pub}$, and a private corpus $\mathcal{D}_{priv}$ (see Algorithm 1). We denote the first three steps—training the domain classifier, clustering, and adjusted importance sampling—as the data selection procedure $M_{\text{sel}} : \mathcal{D}_{pub} \rightarrow \mathcal{D}'_{\text{pub}}$, which operates under privacy budget $(\varepsilon_1, \delta_1)$. The subsequent knowledge distillation step $M_{\text{KD}}$ is a *post-processing* operation on $\mathcal{D}'_{\text{pub}}$, incurring no

---

**Algorithm 1** The PRIVDISTIL Pipeline.

---

**Input:** Public pre-trained LLM, public corpus $\mathcal{D}_{pub}$, private dataset $\mathcal{D}_{priv}$, number of clusters $k$, number of samples $m$, private fine-tuning rounds $R_{priv}$, public knowledge distillation rounds $R_{pub}$, domain classifier training rounds $R_{dc}$, privacy budget $(\varepsilon, \delta)$.
**Output:** Compressed language model satisfying $(\varepsilon, \delta)$-DP.

1: Determine how to split $(\varepsilon, \delta)$ into $(\varepsilon_1, \delta_1)$ for domain classifier training and $(\varepsilon_2, \delta_2)$ for the final fine-tuning
2: *// 1. Domain Classifier Training*
3: Initialize domain classifier $f$
4: **for** $i = 1$ to $R_{dc}$ **do**
5:     Train $f$ with DP-SGD on $\mathcal{D}_{priv} \cup \mathcal{D}_{pub}$ using $(\varepsilon_1, \delta_1)$
6: **end for**
7: *// 2. Public Corpus Clustering*
8: Generate embeddings for samples in $\mathcal{D}_{pub}$ using the LLM
9: Perform $k$-means clustering on these embeddings to obtain $k$ clusters
10: *// 3. Adjusted Importance Sampling*
11: selected_samples $\leftarrow \{\}$
12: **while** —selected_samples— $< m$ **do**
13:     **for** each cluster $C_i$ in the $k$ clusters **do**
14:         $s^* \leftarrow \arg\max_{s \in C_i} f(s)$
15:         Add $s^*$ to selected_samples; remove $s^*$ from $C_i$
16:         **if** —selected_samples— $\geq m$ **then**
17:             **break**
18:         **end if**
19:     **end for**
20: **end while**
21: *// 4. Public Knowledge Distillation*
22: Initialize student model by copying and reducing layers from the LLM
23: **for** $i = 1$ to $R_{pub}$ **do**
24:     Distill knowledge from the LLM on selected_samples
25: **end for**
26: *// 5. Final Fine-tuning*
27: **for** $i = 1$ to $R_{priv}$ **do**
28:     Fine-tune the compressed model on $\mathcal{D}_{priv}$ with DP-SGD using $(\varepsilon_2, \delta_2)$
29: **end for**
30: **return** Compressed language model with DP guarantee

---

additional privacy cost. We then obtain a compressed model

$$\text{LM} \leftarrow M_{\text{KD}}\big(\text{LLM}, \mathcal{D}'_{pub}\big).$$

Finally, LM is fine-tuned on $\mathcal{D}_{priv}$ via a DP algorithm $M_{\text{FT}}$ using a budget of $(\varepsilon_2, \delta_2)$. Combining these steps under the advanced composition property (Steinke, 2022) ensures the entire procedure remains $(\varepsilon, \delta)$-DP:

**Theorem 1** (Advanced Composition of Approximate $(\varepsilon, \delta)$-DP (Steinke, 2022)). *Let*

$$M(\mathcal{D}_{priv}) = \Big(M_{sel}(\mathcal{D}_{priv}),\ M_{KD+FT}\big(\mathcal{D}_{priv}, \mathcal{D}'_{pub}\big)\Big).$$

*Then $M$ is $(\varepsilon, \delta)$-DP relative to $\mathcal{D}_{priv}$ for any $\delta > \delta_1 + \delta_2$, with*

$$\varepsilon = \min\Big\{\varepsilon_1 + \varepsilon_2,\ \tfrac{1}{2}\big(\varepsilon_1^2 + \varepsilon_2^2\big) + \sqrt{2\log\big(1/\delta'\big)\big(\varepsilon_1^2 + \varepsilon_2^2\big)}\Big\},$$

*where $\delta' = \delta - (\delta_1 + \delta_2)$.*

In our experiments, we use the *private random variable* (PRV) accountant (Gopi et al., 2021; Ghazi et al., 2022) to obtain tighter numerical bounds. Other advanced accountants, such as Rényi Differential Privacy (RDP) (Mironov, 2017), could be similarly applied to track and manage the overall privacy cost.

## B  TRAINING AND HYPERPARAMETER DETAILS

For both domain classifier training and model compression, we follow the same configurations across all datasets, as described in Section 4.1. Note that each sample in MNLI, QNLI, and QQP

includes two separate sentences. To integrate these into a mixed dataset with the public corpus, we handle each sentence independently. Consequently, we train two domain classifiers for each dataset and select the one with the higher prediction loss. Specifically, we use the "premise" field for MNLI, the "sentence" field for QNLI, and the "answer" field for QQP. Compared to DPKD (Mireshghallah et al., 2022), our setup employs fewer total epochs (3 for domain classifier training, 10 for model compression, and 20 for fine-tuning, versus 75 in DPKD) and larger batch sizes during fine-tuning. We implement these large batches efficiently using Opacus's *BatchMemoryManager*.

Table 5: Detailed hyperparameter settings.

| Dataset | Size | $\delta(\varepsilon = 4/\varepsilon = 1)$ | Learning Rate | Batch Size |
|---------|------|------|------|------|
| SST2 | 67K | 1.5e-5 / 1.5e-6 | 5e-4 | 2048 |
| QNLI | 105K | 9.5e-6 / 9.5e-7 | 8e-4 | 8192 |
| MNLI | 393K | 2.5e-6 / 2.5e-7 | 8e-4 | 16384 |
| QQP | 364K | 2.7e-6 / 2.7e-7 | 5e-4 | 8192 |

**Domain Classifier Training.** We set the number of epochs to 3, batch size to 1,024, and learning rate to 0.001. We randomly select 50k private and 200k public samples to fine-tune the classifier.

**Data Selection.** We vary the number of clusters $k \in \{1, 2, 3, 5, 8, 10, 15, 20\}$ and select an equal number of samples from each cluster, with the total number of selected samples in $\{200k, 500k, 1M, 2M\}$. By default, $k = 10$ and we select 500k samples in total.

**Model Compression.** We train for 10 epochs with a batch size of 800 and a learning rate of 0.0005. The distillation loss combines multiple objectives, each given equal weight. To improve performance, we initialize the student model with specific layers from the teacher, following Distil-BERT (Sanh et al., 2019).

**Fine-tuning.** We fine-tune for 20 epochs, varying the learning rate in $\{5.0 \times 10^{-4}, 8.0 \times 10^{-4}, 1.0 \times 10^{-3}, 1.2 \times 10^{-3}, 2.0 \times 10^{-3}\}$ and batch size in $\{2048, 4096, 8192, 16384\}$. We report the best result from all hyperparameter combinations. As is standard practice (Mireshghallah et al., 2022; Tramer & Boneh, 2020), we do not account for hyperparameter tuning in the privacy budget.

**Implementations.** We use Opacus 1.4.0, Transformers 4.31.0, PyTorch 2.0.1, scikit-learn 1.2.2, and Python 3.8.16. Our experiments run on Tesla V100 GPUs. We use distributed training in PyTorch for model compression, and single-GPU training for DP domain classifier training and DP fine-tuning.

## C    IMPACT OF MODEL ARCHITECTURE

In this experiment, we investigate how model architecture affects PRIVDISTIL. We evaluate two teacher models: BERT (Devlin et al., 2019) and GPT-2 (Radford et al., 2019). For GPT-2, we examine the compression process from GPT-2 to DistilGPT2. The private dataset used in this experiment is fixed as SST2.

**Observations**. The results, illustrated in Table 6, demonstrate the influence of model architecture on the effectiveness of PRIVDISTIL. Overall, PRIVDISTIL is highly effective and generally outperforms DPKD in both configurations, highlighting its adaptability and stability.

Moreover, we observe that the effectiveness of PRIVDISTIL does not significantly vary across different model architectures. Consequently, in the subsequent sections, we will use BERT as the fixed model architecture for a more in-depth investigation of our approach.

## D    IMPACT OF CLUSTERING METHODS

Recall that we utilize the embeddings generated by the teacher model to execute clustering, thereby potentially leveraging the knowledge of the teacher model and consequently achieving proficient performance. To understand the impact of the clustering method, we compare the proposed clustering method with a randomly implemented clustering method, which is defined as follows. Suppose

Table 6: The impact of model architecture.

| Teacher | Training | $\varepsilon = 1$ | $\varepsilon = 4$ |
|---------|----------|-------------------|-------------------|
| BERT | DPKD | 78.5 | 82.7 |
| | PRIVDISTIL (Ours) | **85.6** | **89.2** |
| GPT-2 | DPKD | 80.2 | 85.3 |
| | PRIVDISTIL (Ours) | **84.4** | **86.8** |

$\mathcal{D}_{pub}$ is our data and $k$ is the number of clusters we aim to create. An evenly random clustering method can be described as:

$$C_i = \{x \in \mathcal{D}_{pub} : x \mod k = i\}$$

In this definition, $x$ represents an individual data point, and $i$ symbolizes a specific cluster index. Each data point $x$ is assigned to a cluster $C_i$ based on the remainder of the division of $x$ by $k$. It should be noted that random clustering can also control over generalization ability since conducting adjusted importance sampling on the generated clusters can also enhance the diversity of the selected data. However, it does not utilize any knowledge from the teacher model.

In this experiment, we fix the private dataset as SST2, set the privacy budget to $\varepsilon = 4$, fix the teacher model to $\text{BERT}_{\text{BASE}}$, and maintain a compression ratio of 0.5.

**Observations**. Figure 5 compares the end-to-end performance between the random clustering method and the proposed method, which uses the embedding generated by the teacher model. We can observe that random clustering typically achieves peak performance when the the number of clusters is 15, with an accuracy of 88.7%. On the other hand, the proposed method reaches peak performance when the number of clusters is 10, attaining an accuracy of 89.2%. Therefore, the peak performance achieved by the proposed method is considerably higher than that of random clustering. This highlights the importance of using the knowledge extracted from the teacher model.

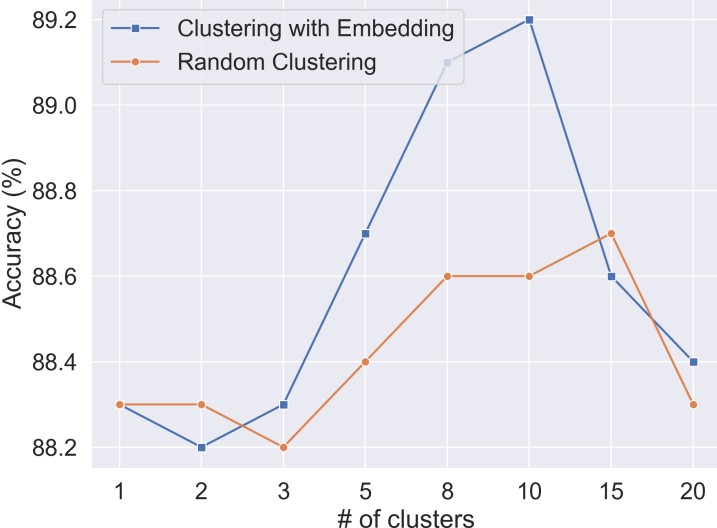

Figure 5: The impact of clustering method.

## E IMPACT OF INITIALIZATION METHOD

Recall that our knowledge distillation method aligns with both DistilBERT (Sanh et al., 2019) and DPKD (Mireshghallah et al., 2022), initializing the student network using the teacher model's

weights. In the proposed PRIVDISTIL framework, compression occurs before fine-tuning; thus, the compressed model can only be initialized using the pre-trained teacher model's weights. Conversely, in DPKD, fine-tuning occurs before compression, providing an alternative option to initialize using the weights of the fine-tuned teacher model. This section offers a comparative analysis between PRIVDISTIL and both versions of DPKD, with initialization using the pre-trained teacher model denoted as PT, and using the fine-tuned teacher model as FT. In this experiment, we set the privacy budget to $\varepsilon = 4$, fix the teacher model to $\text{BERT}_{\text{BASE}}$, and maintain a compression ratio of 0.5.

**Observations**. Table 7 illustrates the impact of the initialization method on the performance of DPKD. We observe that when $\varepsilon = 4$, utilizing weights from the pre-trained teacher model yields superior results, whereas when $\varepsilon = 1$, employing weights from the fine-tuned teacher model is more effective. However, there is only a modest average variance of only $\pm 0.3\%$. Nevertheless, the proposed PRIVDISTIL framework outperforms both versions of DPKD across all tested privacy budget scenarios.

Table 7: The impact of the initialization method.

| Budget | Training | Init. | MNLI | QNLI | QQP | SST2 | Avg |
|---|---|---|---|---|---|---|---|
| $\varepsilon = 1$ | DPKD | FT | 68.3 | 80.3 | 77.0 | 80.0 | 76.4 |
| | | PT | 67.5 | 80.1 | 78.4 | 78.5 | 76.1 |
| | PRIVDISTIL (Ours) | PT | **68.5** | **80.1** | **82.3** | **86.7** | **79.4** |
| $\varepsilon = 4$ | DPKD | FT | 72.3 | 82.9 | 82.1 | 82.6 | 80.0 |
| | | PT | 72.8 | 83.0 | 82.6 | 82.7 | 80.3 |
| | PRIVDISTIL (Ours) | PT | **73.3** | **83.3** | **84.8** | **89.2** | **82.7** |

## F    ERROR RATE OF DOMAIN CLASSIFIERS

In this section, we report the performance of our domain classifiers. For each private dataset, we randomly select 50,000 samples from the private corpus and 250,000 samples from the public corpus to fine-tune the domain classifier. We then measure the error rate (i.e., the misclassification rate) on a test set composed of 15,000 private examples and 15,000 public examples. Throughout this section, the entire privacy budget is allocated solely to fine-tuning the domain classifier, so the $\varepsilon = 1$ setting here corresponds to the same setting used in our main text. Table 8 summarizes the results. We observe that DP noise has a minimal impact on the classifier's performance; the classifier can accurately distinguish between private and public examples. We attribute this to the simplicity of the domain classification task in our scenario and the relatively large gap between private and public data distributions.

Table 8: The error rate of domain classifiers.

| | SST2 | QQP | QNLI | MNLI |
|---|---|---|---|---|
| $\varepsilon = 1$ | 0 | 1.7e-5 | 3.7e-5 | 6.0e-5 |
| $\varepsilon = 2$ | 0 | 1.3e-5 | 3.7e-5 | 2.0e-5 |
| $\varepsilon = 4$ | 0 | 1.3e-5 | 2.3e-5 | 2.0e-5 |

## G    DISCUSSION

PRIVDISTIL is designed with data selection as its principal component to enhance the privacy-utility trade-off. However, this framework relies on data selected from a provided public corpus; thus, in the absence of public data for specific scenarios—such as some medical tasks—it cannot function. In such cases, the domain classifier, originally used as a sampler for data selection, can be replaced with a DP generator. This generator, which might be a diffusion model, can be trained on private data to produce synthetic public data that mirrors the private data's distribution. The newly generated

public data then serves as training material for the knowledge distillation process. Future research could explore this approach further.

Notably, the requirement for public data (auxiliary data) is a common assumption in the field of DP machine learning research. This assumption is not unique to our paper but is also present in methods such as PATE (Papernot et al., 2017) and Private-knn (Zhu et al., 2020). Auxiliary data generally refers to data that share the same distribution as sensitive data but is publicly available. Nonetheless, we have demonstrated that our strategy exhibits good transferability and is not strictly limited to public datasets with the same distribution. We believe this represents a relaxation of the auxiliary data assumption compared to prior works in DP machine learning.

Furthermore, while our discussion and implementation center on a single-party scenario, it is worth noting that our framework is adaptable to both single-party and multi-party situations. In a multi-party context, the private data originates from multiple sources. Consequently, both the domain classifier training and private fine-tuning processes can be facilitated through federated learning. In these instances, the DP-SGD algorithm would need to be supplanted by the DP-FedAvg algorithm (McMahan et al., 2017) to integrate DP noise during aggregation. Although PRIVDISTIL provides a promising foundation, the adaptation, execution, and assessment of the methodology in multi-party contexts necessitate further exploration and rigorous validation. Future research endeavors will delve deeper into the complexities and nuances of multi-party scenarios, evaluating the adaptability, scalability, and efficiency of PRIVDISTIL in these environments. We also note that the proposed PRIVDISTIL can be further combined with pruning strategies Gordon et al. (2020); Hou et al. (2020) and quantization strategies for enhanced efficiency Shen et al. (2020); Zafrir et al. (2019). We leave the exploration of this combination to future work. PRIVDISTIL is also considered suitable for the image domain. However, this application becomes more challenging since it is much harder to find similar data samples in the image domain compared to the language domain. We leave this exploration to future work.

