# OpenReview forum: "PRIVDISTIL: A Unified Framework for Accurate and Differentially Private Model Compression"
_ICLR.cc/2026/Conference — Submitted to ICLR 2026_

### Official Review · Reviewer_TbmF · 2025-11-01

**Soundness:** 3
**Presentation:** 3
**Contribution:** 2
**Rating:** 4
**Confidence:** 4

**Summary:**

Authors present PRIVDISTIL, a differential privacy (DP) aware model compression technique. Primary challenge that authors address is the following: a framework that compresses models without major regressions to task quality while preserving privacy. They do so by primarily distilling on "selected public data" and then eventually doing DP-SGD on the student model. The key to this process is data selection which is called importance sampling and ensuring diversity through clustering. Authors show that they have substantial gain in performance 3% gain in GLUE benchmark under a tight privacy budget ε= 1. The work is empirical, and while the technique does not seem highly novel appears efficient when compared to existing DP based compression techniques.

**Strengths:**

1. Clean and clear framework: the design is modular easy to implement, scale and has practical applicability. DP Domain Classifier -> Public Corpus Clustering -> Adjusted Importance Sampling -> Public KD followed by DPSGD on private data. The authors have explained each step and overall intuition behind each process in a detailed manner

2. Strong empirical results: Compared to DPKD shows strong gains especially at smaller privacy budget
BERTBASE/LARGE and ε∈{1,4}, PRIVDISTIL improves mean GLUE accuracy (79.2 vs. 76.1 at ε=1 with BERTBASE; 77.9 vs 75.9 )

**Weaknesses:**

1. Ablations and evals can be more rigorous: While there are tables on the impact of clusters and data selection, it is not apparent which part of the process is the giving the primary lift. For ex: AIS improves +0.8 avg compared to full data but 7.0 over random. It appears the major gains are coming from public KD

2. Stat significance: All tables appear to be single run only. Seeing confidence intervals with difference seeds can provide more evidence

3. While clipping params and epsilon 0.3 are specified noise multipliers, total steps per data set are not specified.

Minor
4. It is not specified what hardware is used for comparing speed ups

Overall, since this is an empirical work (there is precedence around using public data along with private/DP mechanisms like private kNN, PATE) stronger ablations, and experiments are required. It is not clear from the paper what is providing the lift in quality metrics

**Questions:**

1. [Minor] Importance sampling seems to be a bit misleading. This is not importance sampling in classic sense but rather cluster based greedy selectiion, topk. Please clarify.

2. In appendix F  it appears domain classifier has perfect separability with close to zero error rate. In real world scenarios this may not be likely. Is it possible to use more challenging data and present results?

3. Ablations and evals require more rigor as stated above - it is not clear which part of the process is playing a role in improving quality. Please provide more experimentation

4. Need more rigorous measurement of efficiency/speedup - is this latency? If yes under what conditions?

5. [Good to have] While it is great to see results with BERT does this scale with larger models? Discussion would be if not for some preliminary experimentation.

6. Statistically significant results are needed. Good to see results over multiple runs

---

### Official Review · Reviewer_bpHu · 2025-11-01

**Soundness:** 3
**Presentation:** 3
**Contribution:** 2
**Rating:** 4
**Confidence:** 3

**Summary:**

This paper considers the process of fine-tuning and compressing LLMs under differential privacy. Compression is neccessary when LLMs are deployed on memory-constrained devices. The paper shows improved results over a simple baseline of compressing and fine-tuning using DP-SGD (Mirshegallah et al 2022). They propose using public data for the compression process, but instead of using the public data as is, they instead subsample the data to ensure closeness in domain to the private data and diversity within the subsampled data. The subsampling (called Adjusted Importance Subsampling) is their main technical contribution. WIth the access to public data + the subsampling strategy, they show large improvements over the simple baselien for 2 out of 4 tasks (between 3-6%) and smaller improvements (<1%) on the other 2 tasks.

**Strengths:**

- Clear improvement over a simple baseline.

- Comprehensive experiments and ablation studies.

- Results are presented in a clear way.

**Weaknesses:**

- It would be good to see results at larger epsilon values e.g., eps=10, does the method still hold a significant advantage over the DPKD baseline?

- I am still not convinced about the motivation for this problem. If the compressed model is to be deployed on device it is usually becasue it will process very sensitive data, e.g., text messages. Your method requires obtaining such data on the server side to perform the compression+fine-tuning, but collecting very sensitive data on the server side seems challenging. What kind of tasks/data are you considering your method would apply to?

- I am also not clear about the importance of the subsampling from the public corpus, which is a key technical contribution. In Table 4, first row, am I understanding corresctly that this row corresponds to doing compression using the full data and then fine-tuning with the private data? If so, the gains of your data subsampling method over the first row are on the smaller side (1-1.5%) and I assume they would be smaller for larger epsilon. This means that the access to public data, without any data sampling strategy, is a key way in which you obtain the improved results over the DPKD baseline, which does not use public data, thus it seems to me an unfair comparsion to the baseline.

- It is also not clear how performant your method will be when there is a large distribution gap between the public corpus used and the private corpus. If we are unable to obtain updated public data from the private data domain, such a large gap will eventually emerge.

- Finally DPKD is the simplest baseline in Mirshegallah et al, which they acknolwedge is on the weaker side, and they discuss other techinques such as initialization and pruning to increase peformance. Am I correct that in your comparsion you did not use any of their baselines involving pruning and initialization?

- This is more minor, but in Table 1 I am still having a hard time understanding what each of the rows corresponds to (basides the DPKD and PrivDistil rows). I suggest clearly stating what is the teacher and student model and whether is any compression happenning.

**Questions:**

Please see questions in the section above.

---

### Official Review · Reviewer_g2DN · 2025-11-02

**Soundness:** 3
**Presentation:** 4
**Contribution:** 2
**Rating:** 4
**Confidence:** 5

**Summary:**

The paper tackles the challenge of compressing language models under differential privacy (DP) without sacrificing too much utility. The authors propose PRIVDISTIL, (1) trains a DP domain classifier to estimate which public samples resemble the private domain, (2) clusters the public corpus using teacher embeddings, (3) performs Adjusted Importance Sampling (AIS) to pick relevant-and-diverse public data, (4) runs non-private distillation on those public samples to build a compact student, and (5) finally DP-fine-tunes the compressed student on private data. The whole process composes to an (ε,δ)-DP guarantee; in experiments on GLUE, PRIVDISTIL improves accuracy over DPKD at ε∈{1,4} and yields notable speedups via DistilBERT students.

**Strengths:**

Idea is nice: It is good to exploit public data for DP model compression. Training a DP domain classifier to rank public samples aligns selection with the private task while spending only a small slice of the privacy budget; the rest of compression occurs non-privately on selected public data.

Quality and Clarity: The paper is very well written and the idea is clearly presented.  It has clear end-to-end privacy accounting. The paper gives a composition statement and uses a PRV accountant for tighter bounds; it also explains the budget split across domain-classifier training and final DP fine-tuning.

Practical wins under realistic budgets. Gains at ε=1 and ε=4 on standard GLUE tasks, plus speedups from DistilBERT, make the method attractive in practice.

**Weaknesses:**

The big concern is that the paper does not include an important related work that also tackles differential private model compression with public data selection. These two papers share similar idea and [1] seems achieving better utility on MNLI and SST-2 under private budget epsilon=4.

[1] Yu et al., Selective Pre-training for Private Fine-tuning, TMLR 2024

Given the rapid movement of LLM, the paper does not include experiments on many strong autoregressive models, like Qwen3 0.6B or miniCPM4 0.5B.

**Questions:**

See weakness.

---

### Meta-Review · Area_Chair_rEbz · 2026-01-07

**Summary:**

Reviewers concerns are not addressed, including lack of comparisons, missing baselines, and experimental settings. Comparisons to important existing baselines are missing in the experiments. The setting only addresses limited values of epsilon and model sizes, and ablation is weak. At this point, the authors have declined to respond to the reviewer comments, and it is unlikely that the reviewers would have changed their assessment about the paper.

**Reviewer Concerns:**

No reviewer concerns are addressed.

**Reviewer Scores:**

The scores would have remained the same.

---

### Decision · Program_Chairs · 2026-01-26

Reject